# Biphasic Effects of Blue Light Irradiation on Different Drug-Resistant Bacterium and Exploration of Its Mechanism

**DOI:** 10.3390/biomedicines13040868

**Published:** 2025-04-03

**Authors:** Yifei Mu, Yilin Shen, Norbert Gretz, Marielle Bouschbacher, Thomas Miethke, Michael Keese

**Affiliations:** 1Department of Gastrointestinal Surgery, Renji Hospital, Shanghai Jiao Tong University School of Medicine, Shanghai 200127, China; 18076@renji.com; 2Center of Medical Research, Bioinformatics and Statistics, Medical Faculty Mannheim, Heidelberg University, 68167 Mannheim, Germany; norbert.gretz@medma.uni-heidelberg.de; 3Department of Otolaryngology & Head and Neck Surgery, Ruijin Hospital, Shanghai Jiao Tong University School of Medicine, Shanghai 200070, China; syl13026@rjh.com.cn; 4Urgo Research Innovation and Development, 21300 Chenôve, France; m.bouschbacher@fr.urgo.com; 5Institute of Medical Microbiology and Hygiene, Medical Faculty Mannheim, Heidelberg University, 68167 Mannheim, Germany; 6Department of Surgery, Medical Centre Mannheim, Medical Faculty Manheim, Heidelberg University, 68167 Mannheim, Germany; 7European Center of Angioscience (ECAS), Medical Faculty Manheim, Heidelberg University, 68167 Mannheim, Germany; 8Department for Vascular Surgery, Theresienkrankenhaus Mannheim, 68165 Mannheim, Germany

**Keywords:** blue light radiation, antibiotic therapy, constant mode, cycling mode, TCA cycle

## Abstract

**Background**: Antimicrobial resistance is a problem that threatens the entire world population. Blue light irradiation (BLI) is a novel technology with a bactericidal effect. However, it has only been employed in experimental and preclinical trials. **Methods**: We employed BLI on four kinds of bacteria (*Staphylococcus aureus*, *Pseudomonas aeruginosa*, *Proteus mirabilis*, *Klebsiella pneumoniae*, and *Escherichia coli*) and six kinds of artificial implants utilizing a BioLight LED lamp and MEDILIGHT at a 453 nm wavelength. **Results:** The results showed that the antibacterial effect of BLI enhanced with time and dosage. Irradiation of 165.6 J/cm^2^ corresponding to 120 min of constant mode irradiation, resulted in a significant reduction in the CFU for all four strains. Moreover, the cycling mode (30 s on/30 s off) of the MEDILIGHT prototype showed a more effective microbial effect compared to the constant mode using the BioLight LED lamp. *Pseudomonas aeruginosa* was the most sensitive strain to BLI, and *Staphylococcus aureus* showed relatively greater resistance to BLI. BLI showed different antibacterial effects on various types of implants, indicating that different physical properties of artificial implants were more likely to influence the bactericidal effect of BLI. Decreased ATP highlighted energy deprivation after BLI. Genechip analysis of *Escherichia coli* under constant mode and cycling mode of BLI revealed the downregulation of metabolism-related pathways, and most genes involved in the TCA cycle were downregulated. **Conclusions**: Our results showed that cycling mode BLI has great potential for use in future disinfection applications. We also proposed a new viewpoint that energy deprivation might be another possible mechanism underlying the antibacterial effect of BLI.

## 1. Introduction

Antibiotics are the most effective medications to fight against bacterial infections. However, the widespread use of antibiotics induced a marked increase in antimicrobial resistance, which is now threatening this therapeutic accomplishment, jeopardizing the successful outcomes in critically ill patients [1]. It is now indisputable that antibiotic resistance is as life-threatening as cancer, both in terms of cases and the likely outcomes [2]. According to statistics, patients with chronic nonhealing wounds represent approximately 15% of clinical care beneficiaries. Total Medicare spending is estimated to be close to USD 100 billion [3]. Surgical infections are the most common cause of chronic infections, followed by diabetic infections. Chronic nonhealing wounds include venous leg ulcers, diabetic foot ulcers, pressure ulcers, and burns [4]. A study in Essen (Germany) indicated that the majority of bacteria found in chronically infected wounds were *Staphylococcus aureus* (*S. aureus*), *Pseudomonas aeruginosa* (*P. aeruginosa*), *Proteus mirabilis* (*P. mirabilis*), *Klebsiella pneumoniae* (*K. pneumoniae*), and *Escherichia coli* (*E. coli*) [5]. The extensive use of antibiotics is the single most critical factor leading to antibiotic resistance [6]. Consequently, there is a critical need to develop novel approaches to tackle antibiotic resistance.

Recently, studies have indicated that blue light in the spectrum of 400–470 nm demonstrates antimicrobial properties [7]. Blue light irradiation is a novel technology that has only been employed in experimental and preclinical trials. It has gained clinical relevance for treating acne vulgaris, an important dermatological disorder caused predominantly by *Cutibacterium acnes*. In a prospective, single-center, open-label study, Wheeland et al. treated 32 patients with facial acne using a handheld LED lamp (412 nm, 29 J/cm^2^/day). After 8 weeks of treatment, 100% of the subjects considered their overall appearance to have improved [8]. Blue light can also inactivate *Helicobacter pylori*, the primary pathogen that causes atrophic gastritis and peptic ulcers. *H. pylori* is increasingly resistant to antibiotics [9]. Ganz et al. treated 10 patients with blue light (405 nm, 40.5 J/cm^2^) via an optical fiber inserted into an endoscope channel that illuminated a 1 cm diameter area of the gastric mucosa. A significant reduction (91%) in *H. pylori* colonies per gram of tissue was observed in treated sites compared to biopsies from adjacent untreated sites. Some patients demonstrated reductions in bacterial burden, approaching 99% [10].

As an innovation in wound research, the aim of the European project MEDILIGHT is to develop a medical device that uses the proven therapeutic effects of visible light to improve the self-healing process of wounds and to monitor their status during therapy [11]. The aim of this study was to test the photoinactivation effect of 450 nm blue light irradiation on diverse bacterial strains (*E. coli*, *P. aeruginosa*, *S. aureus*, and *K. pneumoniae*) with various parameters (irradiance, fluence, irradiation time) in different modes. This is the first time that we have used blue light in the cycling mode of this blue light and explored its antibiotic effect. However, from a previous study, Grzelak et al. reported that cell culture media could produce light-dependent ROS [12]. Pyruvate counteracts ROS and protects cells from photocytotoxicity [13]. To investigate if the nutrient broth medium for bacteria produces ROS post-irradiation, which could kill bacteria after BLI, pyruvate was added to the nutrient broth agar plates in the preliminary experiment. Furthermore, the bactericidal efficacies of the different illumination devices were compared. Once these effects were described, the clinical application of implants was tested. Additionally, the intracellular molecular biological responses were studied to elucidate the underlying mechanism.

## 2. Materials and Methods

### 2.1. Device

Two light sources, including a BioLight LED lamp (Figure 1A) and MEDILIGHT (Figure 1B), were used in the experiments. The BioLight LED lamp contained a driver (LED driver RN 1396, provided by Relco) and an LED array. Both the driver and the LEDs were calibrated and derived using URGO (Chenôve, France). The wavelength was fixed at 453 nm. The BioLight LED lamp provided constant mode irradiation at a constant intensity and was positioned directly (about 1–2 cm) over the top of the Petri dish (Figure 1C). The MEDILIGHT prototype contained a driver, including a rechargeable battery and the illumination system, which consisted of LEDs, sensors, and a pulse oximeter. The wavelength was fixed at 453 nm. The prototype was controlled by software via Bluetooth, which was used to adjust the intensity of the LEDs and enable different modes, including the cycling mode, to control the temperature. The Petri dish was fixed inside a hole of the same diameter. An E2723 wound dressing developed by URGO RID (Chenôve, France) was placed between the lamp and the Petri dish. The device could be opened easily from the side (Figure 1D–G).

### 2.2. Different Blue Light Irradiation Mode

Constant and cycling modes were used in different experiments to verify their antibiotic effects, which were applied in our previous study [11]. A BioLight LED lamp was used to irradiate bacteria in constant mode for different durations before bacterial seeding. MEDILIGHT was used to irradiate bacteria in the cycling mode. The appropriate cycling mode setting was determined to be 30 s on and 30 s off to avoid overheating. The overall irradiation time of the cycling group was identical to the constant group (so the experimental time of cycling mode was twice that of the constant mode). Each sample was tested in triplicate, and each experiment was repeated three times. During irradiation, the temperature of the lamp was always lower than 30 °C. The temperature was measured using a prototypical chip type temperature measuring sensor. Measurements were taken between the prototype and the Petri dish and beneath the Petri dish during irradiation.

### 2.3. Bacteria Strain and Culture

Altogether, four stains were used for experiments (Table 1). *E. coli (K12)* (DSM 18039) was acquired from the medical microbiology department, Mannheim. The other three strains (*P. aeruginosa*: DSM 1128, *S. aureus*: DSM 799, and *K. pneumoniae*: DSM 789) were acquired from DSMZ (Deutsche Sammlung von Mikroorganismen und Zellkulturen GmbH, Braunschweig, Germany). LB nutrient broth (Merck Chemicals GmbH, Darmstadt, Germany) was used for culturing bacteria (all four strains) at a concentration of 8 g/L. After stirring and autoclaving, the medium agar mixture was poured into 92 × 16 mm Petri dishes and then cooled down under a running laminar flow hood to congeal. A colony from an actively growing culture was picked and inoculated into the nutrient broth media and then stored in a shaker incubator at 37 °C. After overnight incubation, the bacteria culture was diluted to 1:50 using media and incubated again to adjust the bacteria back to the log phase. According to the OD600 value, the culture was diluted to the final working concentration. The plates with the colonies were documented using a flatbed scanner (Canon, Tokyo, Japan).

### 2.4. Blue Light Irradiation (BLI) of Bacterial Suspension

After inoculation, overnight incubation, log phase adjustment, and dilution, the bacteria culture was prepared in PBS. Then, the culture was equally separated into several Petri dishes (35*10 mm) according to the number of the experimental and control groups. Each group was tested in triplicate. During constant mode irradiation, the Petri dish was directly positioned underneath the BioLight LED lamp. During cycling mode irradiation using the MEDILIGHT prototype, the relative position between the LEDs, the wound dressing E2723, and the Petri dish was maintained using the fixation device. Then, the bacteria were seeded on the plates and incubated overnight at 37 °C. Then, the plates with the colonies were documented using a flatbed scanner (Canon, Tokyo, Japan).

### 2.5. Blue Light Irradiation (BLI) on Infected Implants with 23 mW/cm^2^

Six kinds of artificial implants were tested for the experiments to verify the antibacterial effect of BLI. They included a bovine pericardium patch (XenoSure^®®^, Burlington, MA, USA), a catheter (GORE^®®^, Newark, NJ, USA), a double lumen catheter (GamCath^®®^, Melbourne, Australia), a patch (GORE^®®^, Newark, NJ, USA), a vascular graft (GORE^®®^, Newark, NJ, USA), and a silver-coated vascular graft (InterGard^®®^, Rastatt, Germany) (Figure 2). All four bacterial strains were used for graft infection. All six implant types were sterilized. The segments or chips of the implants were co-cultured with the bacteria in a small Petri dish (35*10 mm) for 2 h. Then, the segments or the chips were transferred from the bacterial culture to new, empty Petri dishes. They were separated into three BLI groups, namely constant, cycling, and control, with each group tested three times. They then underwent BLI for 120 min at 23 mW/cm^2^ in constant or cycling mode or were not irradiated (control). Afterward, the segments or chips were smeared and seeded onto the agar plates and transferred into the microbiological incubator overnight, except for *S. aureus*, which was incubated for 48 h. The plates were then scanned and evaluated using a flatbed scanner (Canon, Tokyo, Japan).

### 2.6. Oxidative Stress of E. coli After Blue Light Irradiation (BLI)

To access the intracellular oxidative stress of bacteria after BLI, 2′,7′-Dichlorofluorescein diacetate (DCFDA) dye (Sigma-Aldrich, Munich, Germany) was used to determine the generation of reactive oxygen species (ROS) by *E. coli* as an example [14,15]. The bacteria solution with saline received BLI with 23 mW/cm^2^ under different modes. The solution was then transferred into a black 96-well plate, and 10μM DCFDA was added to each well. After incubation in the dark for 30 min, the fluorescence intensity was recorded at excitation and emission wavelengths of 488 nm and 525 nm, respectively, using a microplate reader (Tecan, Männedorf, Switzerland). As a positive control, bacteria were exposed to a gradient concentration of hydrogen peroxide without irradiation before DCFDA was added.

### 2.7. Vitality Determination with ATP Assay

To investigate the vitality of irradiated bacteria, ATP was further tested in *E. coli* as an example using the BacTiter-Glo™ Microbial Cell Viability Assay (Promega, Madison, WI, USA). BacTiter-Glo™ Reagent was stored at −70 °C and was equilibrated to room temperature before use according to the protocol [16]. *E. coli* was grown in nutrient broth at 37 °C overnight. The overnight culture was diluted 50-fold in fresh broth and then incubated for several hours to reach log phase. Samples of the culture were serially diluted using broth in a 96-well plate. BacTiter-Glo Reagent™ was added equally to the volume of bacteria culture medium present in each well. Then, the contents were mixed briefly on an orbital shaker and incubated for five minutes. The plate was transferred to the microplate reader (Tecan, Männedorf, Switzerland), and luminescence was recorded.

### 2.8. RNA Isolation and Quality Control

Bacterial solutions of *E. coli* after 60 min of irradiation at 23 mW/cm^2^ in both constant and cycling modes, as well as the corresponding controls, were harvested for RNA isolation using the RNeasy Protect Bacteria Mini Kit (Qiagen, Hilden, Germany) according to previous research [17]. The concentration and the RNA purity were determined by measuring the absorbance ratio at A260/280 with acceptable values between 1.7 and 2.1 using the Spark^®®^ 10M microplate reader (Tecan, Männedorf, Switzerland). The Agilent 2100 Bioanalyzer (Agilent Technologies, Santa Clara, CA, USA) was used to assess the RNA integrity via capillary electrophoresis with RNA Integrity Number (RIN) > 7, indicating sufficient RNA quality.

### 2.9. Bioinformatic Analysis of GeneChip

Biotinylated antisense cDNA was produced by means of the GeneChip^®®^ WT Plus Reagent Kit and the GeneChip^®®^ Hybridization, Wash, and Stain Kit (both kits were from Thermo Fisher Scientific, Waltham, MA, USA) from 100 ng of isolated RNA. After fragmentation and labeling in a GeneChip Hybridization Oven 640, cDNA was hybridized to the GeneChip™ *E. coli* Genome 2.0 Array from Affymetrix (Santa Clara, CA, USA) for 16 h at 45 °C. Next, the microarrays were dyed using a GeneChip Fluidics Station 450 (Affymetrix) and scanned with a GeneChip Scanner 3000 (Affymetrix) [18]. The annotation was performed with the Custom CDF, version 21, using ENTREZ-based gene definitions. Fluorescence intensity values were normalized by using quantile normalization and Robust Multi-Array Average (RMA) background correction. A commercial software package, JMP Genomics, version 7.1 (Cary, NC, USA), was used to analyze differential gene expression via one-way ANOVA. The public external database Kyoto Encyclopedia of Genes and Genomes (KEGG) provided various pathways. For pathways and gene analyses, a false positive rate of α = 0.05 with false discovery rate (FDR) correction was defined as the level of significance.

### 2.10. Statistical Analysis

Data were analyzed in GraphPad Prism (version 8.4.3, San Diego, CA, USA). All data analyses were performed using SPSS Statistics (version 21.0). Fold changes were calculated by normalizing the colony counts or colony sizes to corresponding non-irradiated controls. For a normal data distribution verified by the Shapiro–Wilk normality test, one-way ANOVA and unpaired Student’s *t*-test were used. For the null hypothesis H0 rejected or abnormal distribution, the Wilcoxon signed-rank test was applied. All numerical data were presented as mean ± SD. The *p*-value of < 0.05 was considered significant.

## 3. Results

### 3.1. Different Power Densities of BLI Showed Distinct Suppression Effects Against Bacteria

To investigate if the nutrient broth medium for bacteria produces ROS post-irradiation, which could kill bacteria after irradiation, *E. coli* was used in preliminary experiments. The agar plates were irradiated for 60 min at 23 mW/cm^2^ at a dosage of 82.8 J/cm^2^ one day before and directly before seeding the bacteria, respectively. The CFU of *E. coli* undergoing BLI one day before seeding decreased 25.9% without significance, whereas BLI directly before seeding decreased CFU by 55.9% (*p* < 0.01). When pyruvate was added to the nutrient broth agar plates, the antibacterial effect of BLI before seeding was not as evident as that before (Appendix A). Thus, we speculated that the ROS produced using broth medium after BLI could also kill bacteria. As a result, in order to further reduce the inhibitory impact from the medium or the agar plates, PBS was used to wash bacteria and dilute the bacterial culture solution. Irradiation was performed in constant mode at different doses. The planktonic *E. coli* solution was first exposed to a relatively low power density of BLI of 10 mW/cm^2^ with different irradiation times or doses. The CFU of all groups of all doses was reduced in comparison with their corresponding non-irradiated controls. The rate of the decline ranged from 2.4% (12.5 min, 17.25 J/cm^2^) to 40.5% (120 min, 165.6 J/cm^2^). However, no significant difference was observed with the use of any dose (Figure 3A). In order to better suppress bacterial growth, a constant irradiation of 23 mW/cm^2^ was applied. Irradiation with a fluence of 23 mW/cm^2^ on *E. coli* showed a more notable inhibition. At lower irradiation doses (2.5 min, 3.45 J/cm^2^; 7.5 min, 10.35 J/cm^2^; 12.5 min, 17.25 J/cm^2^), no significant decrease in CFU formation was observed. When the bacteria were exposed to higher doses of irradiation (30 min, 41.4 J/cm^2^; 60 min, 82.8 J/cm^2^; 120 min, 165.6 J/cm^2^), more CFU inhibition could be observed. The maximum CFU reduction rate was 56.5% (*p* < 0.01) (Figure 3B). The same experiment was performed on *S. aureus* (Figure 3C), *P. aeruginosa* (Figure 3D), and *K. pneumonia* (Figure 3E). The maximum CFU reduction rates were 63.3% (*p* < 0.01), 86.5% (*p* < 0.0001), and 80.3% (*p* < 0.01), respectively. Overall, different bacteria showed different sensitivity to BLI. Among these four strains involved in this study, *Pseudomonas aeruginosa* was the most sensitive. The antibacterial effect of BLI was enhanced with time and dosage. An irradiation dose of 165.6 J/cm^2^, corresponding to 120 min of irradiation, resulted in a significant reduction in the CFU for all four strains.

### 3.2. Cycling Irradiation Mode Showed a Better Bactericidal Effect

The MEDILIGHT prototype was developed in April 2018 to meet clinical requirements. Meanwhile, the prototype was permitted to switch to cycling mode to avoid overheating. To prove the efficacy and the antibacterial effect of the cycling mode (30 s on-time/30 s off-time) of the prototype, a series of experiments were conducted to compare the cycling mode of the MEDILIGHT device to the constant mode using a BioLight LED lamp. All four strains, *E. coli*, *P. aeruginosa*, *S. aureus*, and *K. pneumoniae*, were used in the experiments. The fluence was the same in both constant mode and cycling mode, at 23 mW/cm^2^. Dosages of 10.35 J/cm^2^, 82.8 J/cm^2^, and 165.6 J/cm^2^, corresponding to 7.5, 60, and 120 min of effective irradiation time, were chosen for the experiments. Fold changes were computed using the ratio of CFU from the irradiation group to the corresponding non-irradiated controls. During irradiation, the maximum temperature was 28.5 °C for constant mode using the BioLight LED lamp, and the MEDILIGHT prototype had a maximal temperature of 36.6 °C.

For lower groups that received a lower dose of irradiation (7.5 min, 10.35 J/cm^2^), no change in the proliferation of *E. coli* was observed in either constant or cycling modes (Figure 4A). At higher irradiation doses, cycling mode demonstrated a more substantial inhibitory effect against *E. coli* than constant mode. The survival rates after an irradiation dose of 82.8 J/cm^2^ were 74% for the constant mode and 11% for the cycling mode (*p* < 0.0001). Doses of 165.6 J/cm^2^ in cycling mode showed a bactericidal effect with no CFU remaining. The constant mode irradiation utilizing a dose of 165.6 J/cm^2^ still resulted in a fold change of 5% (*p* < 0.01) (Figure 4A). The same experiment was performed against *S. aureus* (Figure 4B), *P. aeruginosa* (Figure 4C) and *K. pneumoniae* (Figure 4D). Overall, *S. aureus* showed a relatively greater resistance to BLI and no bactericidal effect was achieved, while the *P. aeruginosa* was the most sensitive strain to BLI among four strains involved. Even at lower irradiation doses, the cycling mode exhibited a greater inhibitory effect than the constant mode. *Klebsiella pneumoniae* showed a similar reactiveness to BLI with *E.coli*.

### 3.3. The Bactericidal Use of BLI on Different Kinds of Implants

BLI was used in in vitro tests conducted on medical vascular implants to evaluate the technique’s bactericidal effect. Six types of implants with different shapes and materials, as well as the above four strains of bacteria, were used in the study. Irradiation was applied in both constant and cycling modes with a fluence of 23 mW/cm^2^ and a dose of 165.6 J/cm^2^ (120 min).

Almost no *E.coli*, *P. aeruginosa*, and *K. pneumonia* colonies were found on the central venous catheter (GORE^®®^), vascular patch material (GORE^®®^), or double lumen catheter (GamCath^®®^) after 120 min of cycling irradiation (Figure 5A,C,F). Almost no *E. coli* colonies were found on vascular grafts (GORE^®®^) in either the constant or cycling groups. In addition, no *P. aeruginosa* colonies were found on vascular grafts (GORE^®®^) after 120 min of cycling irradiation (Figure 5B). No significant bactericidal effect was found for silver-coated vascular grafts (InterGard^®®^) (Figure 5D) or the bovine pericardium patch (XenoSure^®®^) (Figure 5E) under either constant or cycling mode. Overall, in the context of implants, the strain most resistant to BLI remained *S. aureus*. BLI in cycling mode showed a better bactericidal effect in the contexts of central venous catheters (GORE^®®^), vascular patch materials (GORE^®®^), double-lumen catheters (GamCath^®®^), and vascular grafts (GORE^®®^).

### 3.4. Change of ROS Level After BLI

2ʹ,7ʹ-Dichlorofluorescin Diacetate (H2DCFDA) is sensitive to hydrogen peroxide and hydroxyl radicals. Here, *E. coli* was also used as an example. Firstly, we needed to determine the effective concentration of *E. coli* used in the experiments. The no-treatment control was designed using NaCl. We found that the fluorescence intensity of DCFDA increased in response to an increasing bacterial concentration. A bacterial concentration between 10^6^ and 10^7^ CFU/mL was chosen and remained the same in each group (Appendix A). Hydrogen peroxide bacteria were then added to the bacteria as a positive control group. When the hydrogen peroxide concentration was lower than 10^−2^ mM, the fluorescence intensity remained low, with no significant difference compared to the controls containing saline and the dye. The ROS levels increased in parallel with a hydrogen peroxide concentration between 10^−2^ mM and 1 mM. As the concentration continuously increased, the fluorescence intensity decreased (Appendix A). In the experimental group, after BLI at a dose of 10.35 J/cm^2^ (5 min), the fluorescence intensity decreased in both groups. The constant mode exhibited a decrease of 28.6% (*p* < 0.0001), and the cycling mode showed a decrease of 36.2% (*p* < 0.0001) compared to the control group (Figure 6A). At a higher dosage (41.4 J/cm^2^, 30 min), the fluorescence was even lower. The ROS level was reduced to 36.6% (*p* < 0.0001) and 33.5% (*p* < 0.0001) in the constant and cycling groups, respectively, compared to the control group (Figure 6B). The difference between the two modes was not significant.

### 3.5. Change in ATP Level After BLI

Adenosine triphosphate (ATP) was then detected in bacteria after BLI. *E. coli* was also used after BLI at a dose of 10.35 J/cm^2^ (5 min). The ATP concentration remained the same in both the constant and cycling modes, without significance (Figure 7A). Conversely, a higher dosage (41.4 J/cm^2^ for 30 min) resulted in a reduction in ATP. As compared to the controls, constant mode irradiation caused a decrease of 14.8% (*p* < 0.001). This inhibitory effect was further enhanced to 47.0% (*p* < 0.0001) after cycling mode irradiation. The difference between the constant and cycling groups was significant (*p* < 0.0001) (Figure 7B).

### 3.6. Gene Expression Analysis from RNA-Sequencing

To investigate the molecular mechanism of the antibacterial effect of BLI, GSEA was conducted using transcriptome analysis. *E. coli* is one of the most studied bacteria in modern biology and is commonly used as a model organism. With its genome sequence determined, *E. coli* was chosen for RNA isolation after BLI with a dose of 82.8 J/cm^2^ under both constant mode and cycling mode (30 s on/30 s off). To balance antibacterial effects and calorigenesis, a fluence value of 23 mW/cm^2^ was used. A total of 10,208 genes were screened using the Affymetrix *E. coli* Genome 2.0 microarray (Appendix A). After constant mode irradiation, a total of 380 genes were significantly upregulated, and 318 genes were significantly downregulated (*p* < 0.05). Conversely, cycling mode led to more differentially expressed genes. A total of 734 genes were significantly upregulated and 857 genes were significantly downregulated (*p* < 0.05) (Table 2).

A ranked list of differentially expressed genes was used to compute normalized enrichment scores (NESs), referring to the gene sets from the KEGG database. Altogether, 40 pathways were enriched and listed. Constant mode BLI resulted in 25 pathways mainly containing downregulated genes and 15 pathways mainly containing upregulated genes (Figure 8A). Among these, six pathways were significantly downregulated, and only one pathway was significantly upregulated (*p* < 0.05). In the cycling mode groups, 23 pathways were downregulated and 17 pathways were upregulated (Figure 8B). Three pathways were significantly downregulated, and only one pathway was significantly upregulated (*p* < 0.05). Considering the KEGG enrichment analysis, we speculated that the downregulation of metabolism-related pathways might play an important role. In addition, pathways related to energy-consuming biological processes, such as biosynthesis pathways (lipopolysaccharide biosynthesis and flagellar assembly), were found to be downregulated in both constant and cycling modes. Carbon metabolism is the main source of energy for bacteria, and it features glycolysis, pentose phosphate pathway, citrate (TCA) cycle, and other carbon metabolic pathways. Among these, the TCA cycle serves as the “power house” to provide cellular energy by producing ATP, modulating NADH/NADPH homeostasis, and scavenging ROS [19]. Combined with the previous results indicating that ATP was reduced after a higher dosage (41.4 J/cm^2^ for 30 min) of BLI, we therefore focused on the genes related to the TCA cycle. Out of 29 genes, altogether, 18 genes showed downregulation trends in both the constant and cycling groups. Only six genes were upregulated in both groups. The expressions of AcnB, icd, fumA, and fumC showed a significant decrease (*p* < 0.05) in the cycling group (Figure 9).

## 4. Discussion

### 4.1. Medium-Irradiance and Cycling Mode BLI Could Have an Improved Bactericidal Effect

Blue light has become a popular research subject, and its antimicrobial effect remains in the exploration stage, especially in the context of its use to battle multidrug-resistant (MDR) bacterial infections [20]. Photocytotoxicity is related to both the irradiance and the dosage, but these factors can also present side effects. Greater illumination also negatively affects cellular growth. Bonatti et al. showed that a high irradiance is harmful to fibroblasts [21]. Yoshida et al. used an even higher irradiance (250 mW/cm^2^) with a lower dose of irradiation (15 J/cm^2^) on fibroblasts, which resulted in the significant inhibition of metabolic activity and cellular structural changes [22]. In this study, blue light (453 nm) with a low irradiance (10 mW/cm^2^) led to no inhibition of the CFU of *E. coli*. A medium irradiance (23 mW/cm^2^) exhibited significant bactericidal effects against all four bacterial strains, namely *E. coli*, *P. aeruginosa*, *S. aureus*, and *K. pneumoniae*. We used irradiance as defined within the MEDILIGHT project featuring a balance between temperature, antibacterial efficiency, and safety for eukaryotic cells. *Staphylococcus aureus* showed the strongest resistance to BLI, while *Pseudomonas aeruginosa* was the most sensitive strain and was sterilized after receiving 165.6 J/cm^2^ of blue light. The findings were the same as those of the study of Maclea [23] and Huang [24] researchers. Gram properties may not be the main factor affecting bacteria’s susceptibility to visible light [25], and more mechanisms must be explored. Research indicates that blue light at 453 nm is nontoxic up to a fluence of 500 J/cm [26]. In addition, BLI at an irradiance of 23 mW/cm^2^ does not lead to apoptosis or necrosis in human cells [11].

To avoid overheating the bacteria, the MEDILIGHT project features a wearable prototype and a cycling mode. The use of a cycling mode with 30 s on/30 s off is an innovation of this study. Similar to constant irradiation, no effect was observed after low doses of irradiation. For *E. coli*, *P. aeruginosa*, and *K. pneumoniae*, sterilization was achieved after receiving 165.6 J/cm^2^ of irradiation. The lowest fluence led to a significant reduction in CFU lower than that in the constant mode. Even *S. aureus* were significantly inhibited after cycling irradiation at 165.6 J/cm^2^. These results imply that the cycling mode had a better bactericidal efficiency.

Few studies have been conducted using cycling or inconsistent irradiation, and their results are controversial. JB Gillespie et al. compared the efficacy of constant and pulsed blue light irradiation [27]. With varying duty cycles (25%, 50%, 75%, and 100%) and different frequencies (100 Hz, 500 Hz, 1 kHz, 5 kHz, and 10 kHz), BLI demonstrated similar efficacies. Another study by DS Masson-Meyers pointed to the same results as ours, indicating that pulsed blue light resulted in a better antibacterial effect than constant irradiation at the same fluence [28]. A pulsed mode at a 33% duty cycle (light emitted 33% of the time, but off 67% of the time during each pulse cycle at a constant current) at 30 min intervals is a commonly used method and has been shown to inactivate *Propionibacterium acnes* [28] and methicillin-resistant *S. aureus* (MRSA) [29], but *Group B Streptococcus* (GBS) is not susceptible to 450 nm pulsed blue light (PBL) unless the bacterium is co-cultured with exogenous porphyrin [30]. In contrast to the pulsed irradiation modes with a comparatively high frequency (usually over 33 Hz), we used a cycling mode with 30 s on/30 s off because we have verified in our previous study that this mode could effectively reduce heat generation for future applications in the human body and did not show cell toxicity [11]. Therefore, we expected to observe different effects on cell metabolism and survival. Our research provides a new mode of BLI application for bactericidal use that could provide a better sterilization effect. The reasons why cycling mode is superior to constant mode remain unclear. According to our previous results, the bactericidal effect of blue light may depend on its effect on cell metabolism since the ATP level decreased more in the cycling mode group.

### 4.2. BLI May Also Be a Promising Method for Sterilizing Implants

Biomedical implants, such as catheters or vascular grafts, increase the risk of infection. Infection is one of the most frequent and severe complications associated with the use of biomaterials [31]. Current prevention strategies, such as perioperative antibiotics or local antiseptic solutions, have limitations and may even cause new complications. Therefore, an improved and more reliable method is required to prevent and treat implant-related infections. Few studies have explored the application of blue light for bactericidal use for implants, and most were only concerned with single-bacterium experiments. Azizi et al. found that light-emitting diodes (LEDs) emitting toluidine blue light could cause significant reductions in *Prevotella intermedia*, *Actinomyces actinomycetemcomitans*, and *Porphyromonas gingivalis* in 72 zirconia dental implants [32]. Antimicrobial blue light devices managing the biofilm burden at the skin–implant interface of percutaneous osseointegrated implants have been researched and developed. The cooling system and the power outputs required further improvement [33]. In our study, we showed that blue light, which has both antimicrobial and noncytotoxic effects, is a valuable tool. Vascular grafts (GORE^®®^) feature a double wall structure that could hold a small amount of bacterial solution in the interlayer. The silver-coated grafts (InterGard^®®^) and the bovine pericardium patch (XenoSure^®®^) both had excellent hygroscopicity, which could hold a large amount of bacteria. The ability to shield bacteria from BLI is an important factor contributing to the low effectiveness of BLI on these materials. Although the catheter, the graft, and the patch from GORE^®®^ all comprised the same material, different BLI efficiencies were observed. Interestingly, BLI was largely inefficient in the context of the only implant that underwent antibacterial pretreatment, graft silver coating (InterGard^®®^). Hence, physical properties such as structure, shape, reflectivity, roughness, and porosity seem to be the more important factors influencing the bactericidal efficiency of BLI, more important than the material itself. The cycling mode was still more efficient than the constant mode. However, *S. aureus* was still insensitive to cycling mode irradiation at a dose of 165.6 J/cm^2^ on all kinds of implants. As a result, the exploration of more modes of blue light irradiance for sterilization is of considerable value. This technique has broad application prospects.

### 4.3. The Mechanism of BLI on the Inhibition of Bacterial Growth

The first study concerned with the disinfectant properties of light was conducted in the late 19th century [34]. The bactericidal effect was the strongest at the shortest wavelength (UV spectrum) [35]. Currently, UV light is one of the most used disinfectants, especially for air disinfection and surface disinfection. However, constant exposure to UV may increase the risks of skin cancer (for example, basal cell carcinoma, squamous cell carcinoma, and malignant melanoma) [36]. Accidental exposure can also lead to bilateral keratoconjunctivitis and facial erythema, followed by other complications in the skin, eyes, nails, and, eventually, the hair [37]. Unlike UV radiation, visible light is much less harmful. It can be applied without the risk of accidental or intentional illumination of the human tissue [38]. Studies have reported that blue light in the spectrum of 400–470 nm demonstrates antimicrobial properties [7]. The mechanism of bacterial photoinactivation is still not completely understood. A common hypothesis is that blue light excites endogenous photosensitizing chromophores in bacteria, resulting in cytotoxic ROS production [39]. Ramakrishnan et al. reported that after exposure to 162 J/cm^2^ of 405 nm light (15 mW/cm^2^ for 3 h), the ROS levels in *Staphylococcus epidermidis* increased, which was detected by DCFDA [39]. M Galbis-Martínez verified that blue light (408 nm) could induce singlet oxygen in *Myxococcus xanthus* [40]. Blue light with a wavelength of less than 410 nm was used in these studies and demonstrated ultraviolet properties, such as genotoxicity and structural damage to the bacteria. In our study, both a lower (10.35 J/cm^2^) and a higher (41.4 J/cm^2^) dose of irradiation caused a significant decrease in ROS production in *E. coli* using both constant mode and cycling mode without a clear reduction in bacteria. This result contradicts other studies that the generation of ROS by endogenous photosensitizers after BLI could have a bactericidal effect and lead to bacterial death [41,42]. The major source of ROS is oxidative phosphorylation and the electron transport chain during respiration [43]. In this study, gene expression analysis indicated that genes encoding complex II (SdhA, SdhB, SdhC, and SdhD) were all downregulated, which might subsequently block the respiratory chain and explain the reasons for the reduction in ROS levels in *E. coli* after BLI.

ATP is the energy currency in all living organisms. It plays an indispensable role in respiration and metabolism and is the most important energy supplier in many enzymatic reactions [44]. No study has ever explored the ATP levels in bacteria after BLI. In our research, after short-term irradiation (10.35 J/cm^2^), the ATP concentration in *E. coli* remained unchanged after constant mode or cycling mode irradiance. Adversely, after 41.4 J/cm^2^ of irradiation, a significant decrease in ATP was found after the use of both modes. After KEGG analysis, a downregulation in many metabolic pathways was observed, including carbon metabolism. In addition, energy-consuming biological processes such as biosynthesis pathways (lipopolysaccharide biosynthesis and flagellar assembly) were also downregulated after BLI, both under constant and cycling modes. Carbon metabolism is the major source of energy for bacteria and includes glycolysis, pentose phosphate synthesis, the citrate cycle, and other carbon metabolic pathways. The tricarboxylic acid (TCA) cycle is central to energy production and biosynthetic precursor synthesis in aerobic organisms [45].

In mammalian cells, mitochondrial metabolism is linked to the production of ATP and TCA [46]. It is widely believed that blue light modulates the mitochondrial function and oxidative phosphorylation in mammalian cells [47]. Becker et al. reported that ATP levels increased 1 h and 24 h after 5.4 J/cm^2^ of blue light treatment. A higher dose of 21.6 J/cm^2^ BLI led to a decrease in ATP in human keratinocytes [48]. Nakayama et al. found that after 20 min of irradiation at 100 mW/cm^2^ using a blue laser (405 nm), ATP levels decreased in the skin of mice [49]. These observations contradict the findings of Liebmann et al. who described the opposite results in human keratinocytes and skin-derived endothelial cells. In both cell entities, the cellular ATP level was significantly elevated after BLI (453 nm, 100 J/cm2) [26]. However, prokaryotes are structurally and functionally different from eukaryotes [50]. Additionally, prokaryotes have different and more complicated mechanisms for the regulation of aerobic breathing and the production of ATP [51]. KEGG analysis showed that most of the genes involved in the TCA cycle were downregulated. Therefore, we proposed that endogenous ROS levels are not the primary mechanism underlying the blue light sterilization. Instead, the bactericidal effect may be more closely linked to disruptions in bacterial energy metabolism since an obvious reduction of ATP levels was found in our study after BLI.

Aside from carbon metabolism and energy supply, biofilm formation is also critical for infectious bacteria to adapt and survive during environmental changes [52]. From the previous results, it was found that the pathway of biofilm formation was also found to be decreased in both constant and cycling modes, without significance. Biofilms are complex microbial communities encased within a self-produced matrix of extracellular polymeric substances (EPSs) [53]. Previous studies have shown that blue light irradiation has an impact on the bacterial composition and viability of the multispecies biofilms, exerting an inhibitory effect on the signaling pathways associated with their formation [54,55]. This may be attributed to blue light modulating the biosynthesis of EPS components and disrupting the function of bacterial adhesins, which are critical for initial attachment and subsequent biofilm development. As a result, we speculated that for short-term BLI, energy deprivation might contribute more to the inhibition of bacterial growth. More experiments are needed to explore the antibacterial mechanism underlying long-term BLI and the effect of blue light on bacterial biofilms.

### 4.4. Study Limitations

There are several limitations to this study. Firstly, we did not use an illumination device that has both a cycling mode and a constant mode. With such a device, the difference between the two modes will be more convincing. Secondly, we explored a limited number of modes of BLI; more modes should be explored, such as for *S. aureus*, utilizing extended irradiation times or multiple daily treatments to optimize dosage. More multi-drug-resistant strains, as well as the biofilm that defends against bacteria, should also be tested to verify the persistent effect of BLI. Thirdly, the infection of the segments persisted for only 2 h, which prevented biofilm formation, and all irradiation was performed on the surfaces of the implant segments. The exposure methods must be improved, and an infection time of 24–48 h should be tested to explore the effects of BLI on biofilm formation. Fourthly, in vivo animal models and clinical studies were not carried out and considering the cycling mode showed a better antibacterial effect in vitro, this should be completed in the future. BLI showed enormous potential for future applications in disinfection. To apply this method clinically, a highly transparent material will be required to achieve external and internal sterilization. In wound treatment, blue light treatment will allow patients to perform the irradiation themselves, even in an outpatient setting. Long-term irradiation will be of limited use, especially in the operating room.

## 5. Conclusions

Blue light irradiation (453 nm) demonstrated a strong antimicrobial effect, both on agar plates and on planktonic bacteria. The innovative cycling mode of the MEDILIGHT prototype showed a more effective microbial effect than the constant mode of the URGO lamp. We are also the first team to test the effect of blue light irradiation on different medical implants, but the antimicrobial effect varied greatly according to different materials. The ATP decrease and energy deprivation following BLI might be the main mechanisms underlying the inhibition of bacteria. More molecules need to be explored in future studies. All results showed the enormous potential of BLI in future disinfection applications.

## Figures and Tables

**Figure 1 biomedicines-13-00868-f001:**
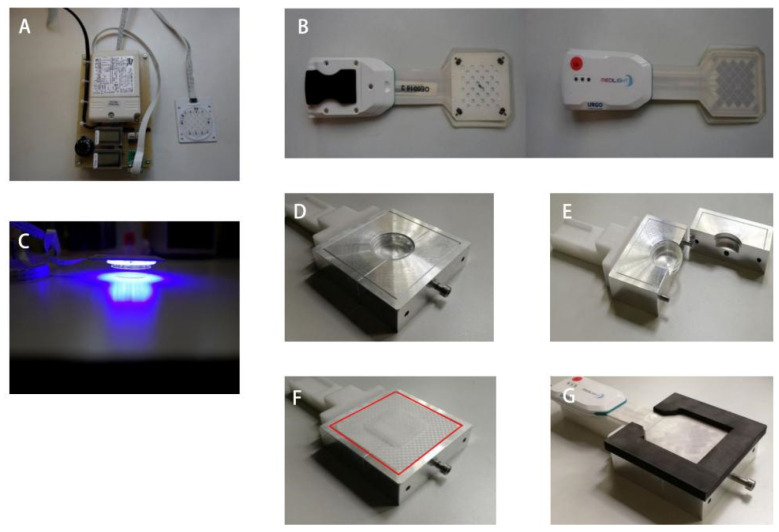
(**A**) The BioLight LED lamp contains the screens for the current temperature as well as the rotary controller for the lamp power. (**B**) The MEDILIGHT prototype consists of the driver and the illumination system from the top side and bottom side. (**C**) Relative position of the BioLight LED lamp and the petri dish during irradiation. The upper bright spot is the BioLight LED lamp and the lower spot is the petri dish. (**D**) The fixation device with the petri dish for the MEDILIGHT prototype. (**E**) The opened fixation device lateral view. (**F**) The groove for the dressing position and the cover board marked with a red line. (**G**) The complete appearance of the fixation device of the MEDILIGHT prototype, including the wound dressing and the petri dish.

**Figure 2 biomedicines-13-00868-f002:**
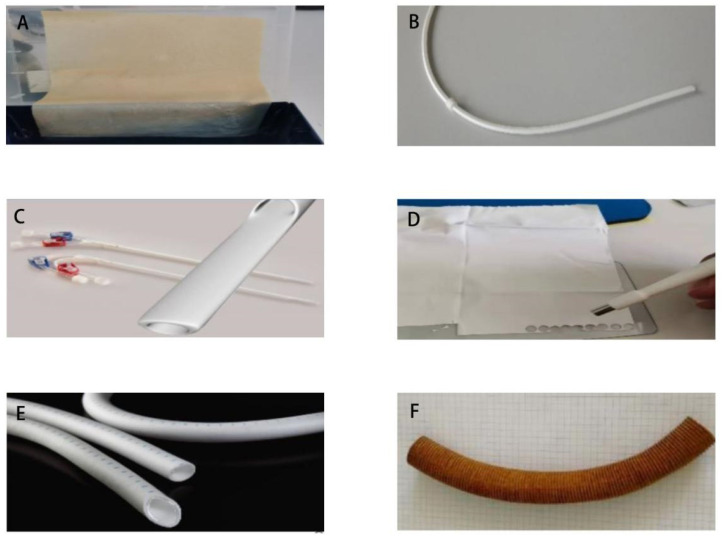
Six kinds of artificial implants. (**A**) Bovine pericardium patch (XenoSure^®®^). (**B**) Catheter (GORE^®®^). (**C**) Double lumen catheter (GamCath^®®^). (**D**) Patch (GORE^®®^). (**E**) Vascular graft (GORE^®®^). (**F**) Silver-coated vascular graft (InterGard^®®^).

**Figure 3 biomedicines-13-00868-f003:**
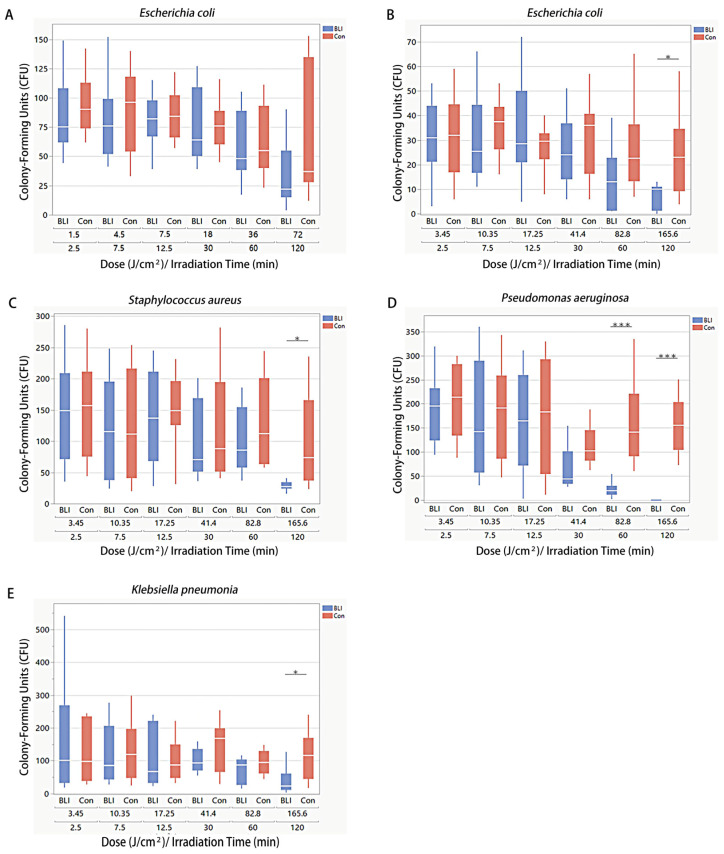
The effect of BLI in constant mode on different planktonic bacteria. (**A**) The effect of BLI on *E. coli* with an irradiance of 10 mW/cm^2^. The effect of BLI on *E. coli* (**B**), *S. aureus* (**C**), *P. aeruginosa* (**D**), *K. pneumoniae* (**E**) with a fluence of 23 mW/cm^2^. Data are presented as box plots with the median, first quartile, and third quartile (N = 4 repetitions, 3 replicates). Data are shown as mean ± SD. “*” represents *p* < 0.01 and “***” represents *p* < 0.0001.

**Figure 4 biomedicines-13-00868-f004:**
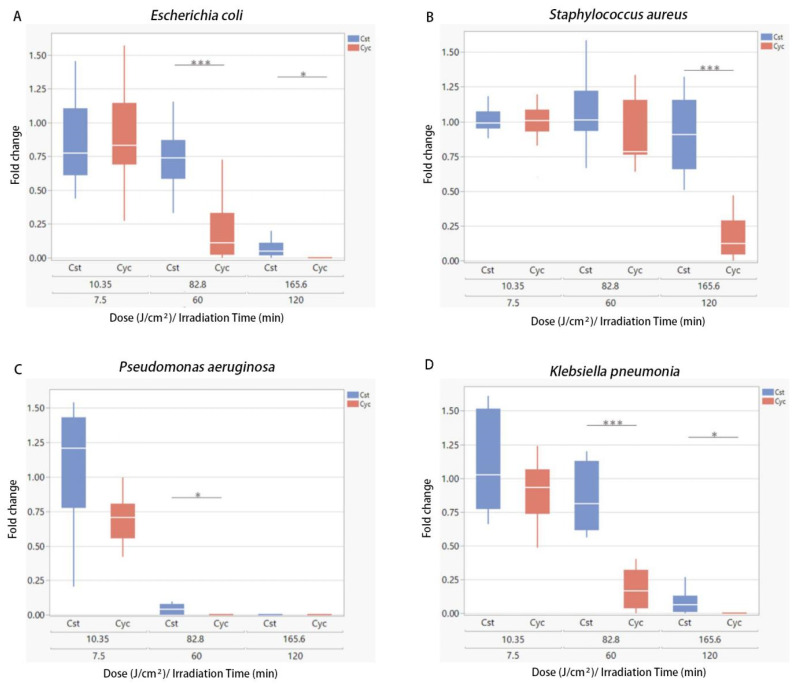
Blue light treatments were performed with an irradiance of 23 mW/cm^2^. Comparison of the antibacterial effect between constant mode (Cst) and cycling mode (Cyc) on *E. coli* (**A**), *S. aureus* (**B**), *P. aeruginosa* (**C**), and *K. pneumoniae* (**D**). Data are presented as box plots with the median, first quartile, and third quartile (N = 3 repetitions, 3 replicates). Data are shown as mean ± SD. “*” represents *p* < 0.01 and “***” represents *p* < 0.0001.

**Figure 5 biomedicines-13-00868-f005:**
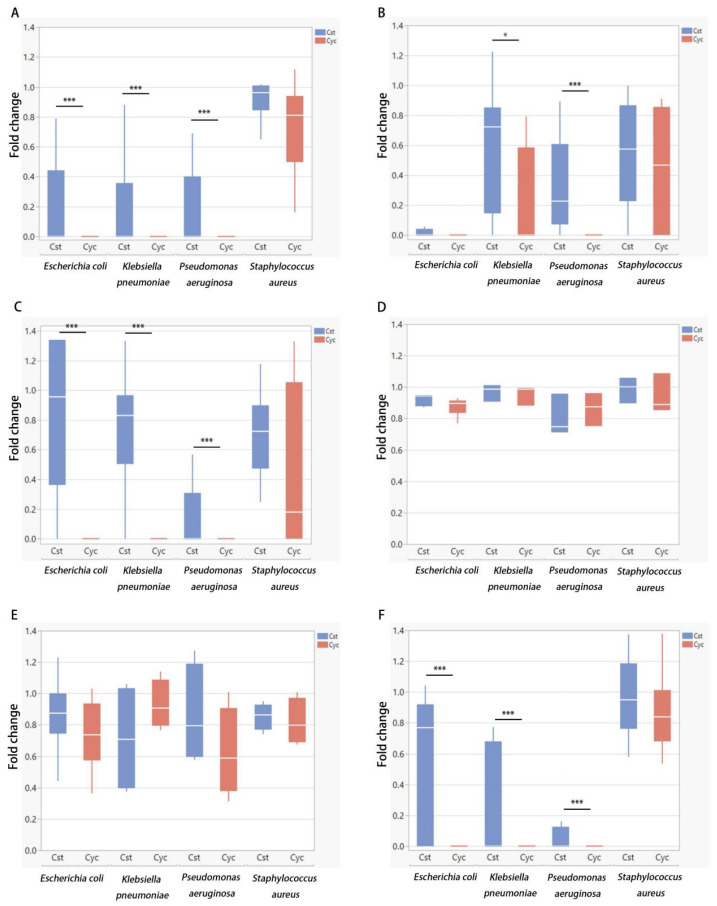
Blue light treatments were performed with an irradiance of 23 mW/cm^2^ and a dose of 165.6 J/cm^2^ (120 min) using both constant mode and cycling mode. Antibacterial effect of irradiation on the central venous catheter (GORE^®®^) (**A**), vascular graft (GORE^®®^) (**B**), vascular patch material (GORE^®®^) (**C**), silver impregnated graft material (InterGard^®®^) (**D**), bovine pericardium patch (XenoSure^®®^) (**E**), double lumen catheter (GamCath^®®^) (**F**). Data are presented as box plots with the median, first quartile, and third quartile (N = 3 repetitions, 3 replicates). Data are shown as mean ± SD. “*” represents *p* < 0.01. “***” represents *p* < 0.0001.

**Figure 6 biomedicines-13-00868-f006:**
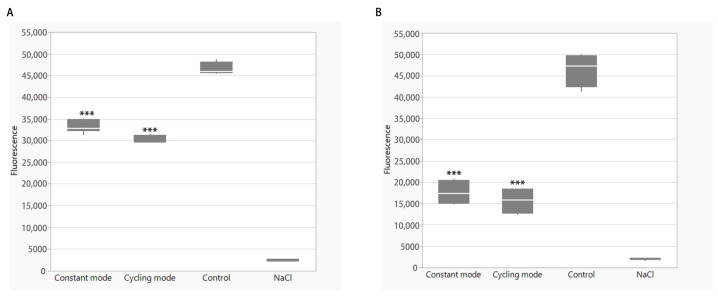
Changes in ROS level of *E. coli* measured by DCFDA after (**A**) 10.35 J/cm^2^ (5 min) irradiation and (**B**) 41.4 J/cm^2^ (30 min). Blue light treatments were performed in both constant mode and cycling mode with an irradiance of 23 mW/cm^2^. Data are presented as box plots with the median, first quartile, and third quartile (N = 3 repetitions, 3 replicates). Data are shown as mean ± SD. “***” represents *p* < 0.0001.

**Figure 7 biomedicines-13-00868-f007:**
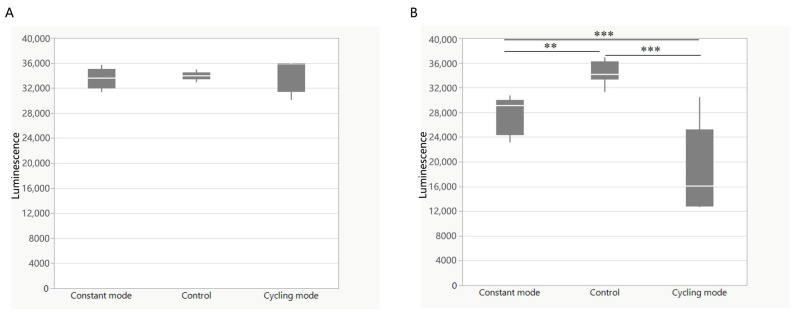
Changes in ATP level of *E. coli* after (**A**) 10.35 J/cm^2^ (5 min) irradiation and (**B**) 41.4 J/cm^2^ (30 min). Blue light treatments were performed in both constant mode and cycling mode with an irradiance of 23 mW/cm^2^. Data are presented as box plots with the median, first quartile, and third quartile (N = 3 repetitions, 3 replicates). Data are shown as mean ± SD. “**” represents *p* < 0.001. “***” represents *p* < 0.0001.

**Figure 8 biomedicines-13-00868-f008:**
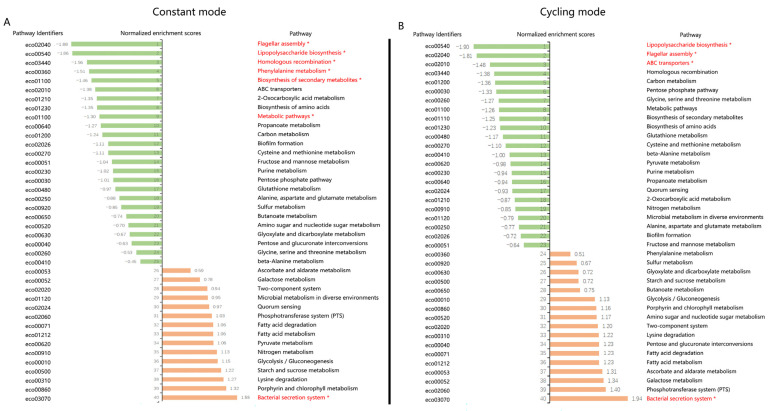
Kyoto Encyclopedia of Genes and Genomes (KEGG) pathway enrichment analysis of RNA sequencing. (**A**) Results of the constant mode versus the non-irradiated control group. (**B**) Results of the cycling mode versus the non-irradiated control group. NES: normalized enrichment score. Pathways marked in red and “*” represents *p* < 0.05.

**Figure 9 biomedicines-13-00868-f009:**
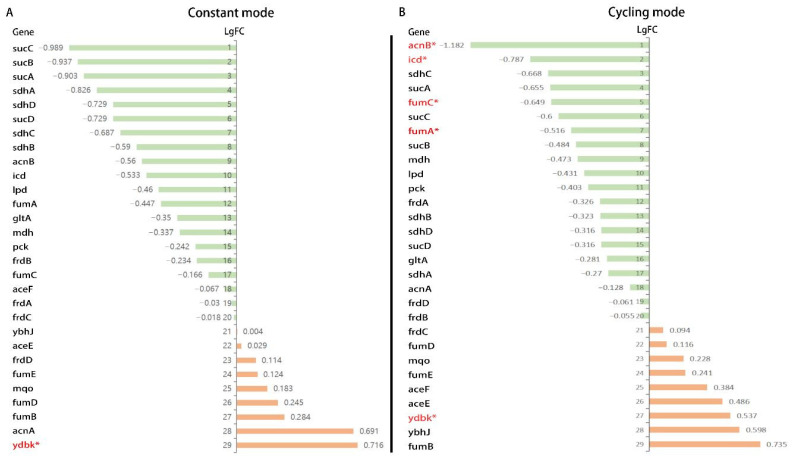
List of genes involved in the KEGG pathway of the citrate cycle. (**A**) Results of the constant mode versus the non-irradiated control group. (**B**) Results of the cycling mode versus the non-irradiated control group. Pathways marked in red and “*” represents *p* < 0.05.

**Table 1 biomedicines-13-00868-t001:** Bacteria strains used for experiments.

Strain	DSM No.
*Escherichia coli (K12)*	DSM 18039
*Pseudomonas aeruginosa*	DSM 1128
*Staphylococcus aureus*	DSM 799
*Klebsiella pneumoniae*	DSM 789

**Table 2 biomedicines-13-00868-t002:** Overview of GSEA displaying the numbers of de-regulated genes.

	Constant Mode vs. Control	Cycling Mode vs. Control
Total genes screened by microarray	10,208	
Significantly upregulated genes	380 (7.48%)	734 (13.27%)
Significantly downregulated genes	318 (6.20%)	857 (18.32%)

## Data Availability

The original contributions presented in this study are included in the article/Appendix A. Further inquiries can be directed to the corresponding authors.

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
