# Peer review of "Biphasic Effects of Blue Light Irradiation on Different Drug-Resistant Bacterium and Exploration of Its Mechanism"

_biomedicines, 2025, doi:10.3390/biomedicines13040868_

Round 1

Reviewer 1 Report

Comments and Suggestions for Authors

Following are my comments and suggestions for improving the manuscript:

  1. Please include some data from the findings in the abstract.

  2. Some sentences in the introduction lack proper references.

  3. Please write out the full name of "MEDILIGHT."

  4. Is there a reference for the methods section 2.2, titled "Different Blue Light Irradiation Mode"?

  5. It would be better to include Table 1 within the text.

  6. Some methodologies are not referenced. This makes it difficult to replicate the experiments.

  7. The resolution of Figure 3 is poor. The axis labels and values need to be larger. The same applies to Figures 4, 5, 6, and 7.

  8. For Table 2, it would be better to include the primer names and sequences.

  9. Consider adding a section discussing the limitations of the study.

  10. The authors should consider investigating the effect on the biofilm and virulence factors of pathogens, instead of focusing solely on carbon metabolism (Fig. 8 and 9).

  11. Provide a section for the abbreviations of each terminology.

Reviewer 2 Report

Comments and Suggestions for Authors

The study presents a well-structured and comprehensive investigation into the antimicrobial potential of blue light irradiation (BLI), specifically at 453 nm, against multiple drug-resistant bacteria and on various medical implants. It compares constant and cycling irradiation modes using two different light sources and explores underlying mechanisms through ROS, ATP levels, and gene expression analysis. However, these points should be improved.
1- The manuscript contains numerous spelling, grammar, and typographical errors (e.g., "be er" instead of "better", "de- creased" instead of "decreased"). These reduce the readability and professional tone of the paper.
2- All experiments were conducted in vitro, and there is no in vivo animal model or clinical study to support the findings' translational value.
3- The explanation of ROS involvement is speculative. Although ROS was measured using DCFDA, it only indicates total oxidative stress—not specific ROS species (e.g., H2O2, O2−, OH•).
The reduction in ROS after BLI is unusual and not well-explained.
4-  The study lacks assessment of bacterial biofilm formation, which is crucial in chronic and implant-associated infections. Biofilms are more resistant to antimicrobial strategies, and this is a major limitation.
5- The implants were infected for only 2 hours before BLI, which is too short to mimic actual clinical infection scenarios where biofilms typically form over 24–48 hours.
6- Although transcriptome analysis shows downregulation of TCA-related genes, no functional validation (e.g., enzyme activity, mutant strains) was conducted to confirm this mechanism.
7- There is no detailed analysis of the physical/chemical properties (e.g., reflectivity, roughness, porosity) of implants, which could significantly affect BLI penetration and efficiency.
8- The term “cycling mode” is introduced as novel, but there is limited comparative discussion with other existing pulsed-light studies. Also, no mechanistic rationale is provided for why cycling mode should be superior to constant mode beyond temperature control.
9- While some p-values are presented, confidence intervals, effect sizes, and detailed statistical outputs are not consistently shown. In some places, it's unclear whether differences are statistically or biologically meaningful.
10- Although a limitations section exists, it does not critically address the key methodological gaps, such as lack of long-term exposure studies, dose optimization, or strain-specific responses.
11- Different devices were used for constant (BioLight Lamp) and cycling (MEDILIGHT) modes. This makes it difficult to isolate the effect of irradiation mode vs. device differences.
12- The paper claims “no resistance developed,” but this conclusion is not supported by long-term or repeated exposure experiments to assess bacterial adaptation.
13- Several ideas are repeated across abstract, introduction, and discussion without adding new insights.
14- Only four bacterial strains were tested. These are common pathogens, but the study would benefit from including multi-drug resistant clinical isolates or anaerobic bacteria.

Comments on the Quality of English Language

English should be improved

Reviewer 3 Report

Comments and Suggestions for Authors

Authors presented the work “Biphasic Effects of Blue Light Irradiation on Different Drug-resistant Bacterium and exploration of its mechanism”. The work is clear and relevant. It is well-written. The introduction provides the context and relevance. The methodological strategy is well documented, including details in all analysis steps.

Overall, the introduction is well-written, providing a solid justification and general context. The relevance of the topic—focused on the use of BLI to influence the bacterial growth of strains of interest—is clearly presented. Additionally, the evaluation following exposure to the proposed techniques is thorough.

The use of scientific names is generally correct, but the simplified format should be standardized throughout the text. In some sections, the full genus and species name is used, while in others, the abbreviated format (first letter of the genus followed by the species name) is applied. This inconsistency should be corrected for uniformity. Additionally, in Figure 5, scientific names should be italicized to align with standard notation conventions.

Some introductory parts of the results section justify certain methodological choices or the overall approach using references. These justifications should be moved to the introduction, with references made to them in the discussion for a more integrated analysis. Such content should not be included in the results section, which should focus solely on the direct implementation of the approach. Please revise these sections accordingly.

In Table 2, please remove figures related to genes with an expression level classified as up or downregulated but that are not statistically significant. Simply reporting the number of genes with expression values above or below a certain threshold is not meaningful unless statistical criteria such as fold change and the corresponding statistical test are applied. Therefore, only differentially expressed genes that meet statistical significance criteria should be reported, as including other values may lead to confusion.

In the methodology section, most aspects are well described. However, it is unclear why ATP measurement, for instance, was performed and reported only for Escherichia coli, while other organisms were included in the study.

Also, for the computational analysis, it is necessary to include software version details, though this is not a major issue. The statistical methods appear robust and appropriate for the type of tests and assumptions made about the analyzed data.

The conclusions section includes study limitations. However, it is recommended that limitations be explicitly stated at the end of the discussion section, integrating them as part of the comparison with other approaches and providing the necessary context for interpreting the results.

Other minor changes:

Italics in line 181.

Line 210: unclear.

Figure 5: Italics required for scientific names.

Table 2: eliminate up regulated and down regulated genes… only significant must be reported. Other are not really up/down regulated. Add % in each case of significant genes.

Round 2

Reviewer 1 Report

Comments and Suggestions for Authors

Well done

Reviewer 2 Report

Comments and Suggestions for Authors

The manuscript was revised as required and could be accepted in its present form